# Tailoring the Radionuclide Encapsulation and Surface Chemistry of La(²²³Ra)VO₄ Nanoparticles for Targeted Alpha Therapy

Miguel Toro-González [1], Allison Peacock [1], Andrew Miskowiec [2], David A. Cullen [3], Roy Copping [1], Saed Mirzadeh [1] and Sandra M. Davern [1,*]

1   Radioisotope Science and Technology Division, Oak Ridge National Laboratory, Oak Ridge, TN 37830, USA; torogonzalmt@ornl.gov (M.T.-G.); owensac@ornl.gov (A.P.); coppingr@ornl.gov (R.C.); mirzadehs@ornl.gov (S.M.)
2   Nuclear Nonproliferation Division, Oak Ridge National Laboratory, Oak Ridge, TN 37830, USA; miskowiecaj@ornl.gov
3   Center for Nanophase Materials Sciences Division, Oak Ridge National Laboratory, Oak Ridge, TN 37830, USA; cullenda@ornl.gov
*   Correspondence: davernsm@ornl.gov

**Abstract:** The development of targeted alpha therapy (TAT) as a viable cancer treatment requires innovative solutions to challenges associated with radionuclide retention to enhance local tumor cytotoxicity and to minimize off-target effects. Nanoparticles (NPs) with high encapsulation and high retention of radionuclides have shown potential in overcoming these issues. This article shows the influence of pH on the structure of lanthanum vanadate (LaVO₄) NPs and its impact on the radiochemical yield of ²²³Ra and subsequent retention of its decay daughters, ²¹¹Pb and ²¹¹Bi. An acidic pH (4.9) results in a high fraction of La(²²³Ra)VO₄ NPs with tetragonal structure (44.6–66.1%) and a ²²³Ra radiochemical yield <40%. Adjusting the pH to 11 yields >80% of La(²²³Ra)VO₄ NPs with monoclinic structure and increases the ²²³Ra radiochemical yield >85%. The leakage of decay daughters from La(²²³Ra)VO₄ NPs (pH 11) was <5% and <0.5% when exposed to deionized water and phosphate-buffered saline, respectively. Altering the surface chemistry of La(²²³Ra)VO₄ NPs with carboxylate and phosphate compounds resulted in a threefold decrease in hydrodynamic diameter and a ²²³Ra radiochemical yield between 74.7% and 99.6%. These results show the importance of tailoring the synthesis parameters and surface chemistry of LaVO₄ NPs to obtain high encapsulation and retention of radionuclides.

**Keywords:** targeted alpha therapy; lanthanum vanadate; radium-223; nanoparticles; functionalization

## 1. Introduction

Targeted alpha therapy (TAT) has been shown to be highly effective in treating small solid tumors and micrometastases, due to the cytotoxic potential of α-particles, which are characterized by a high linear energy transfer (80–100 keV/μm) and short-range in tissue (40–90 μm) [1–3]. In the clinic, current approaches to delivering α-emitting radionuclides involve radionuclide accumulation at the tumor site based on its chemical affinity and the development of radioimmunoconjugates for specific targeting of tumor cells [1,3,4]. Although the accumulation of radionuclides based on their chemical affinity for specific organs and tissues has been successful in treating cancer, this approach limits the potential of delivering therapeutic radionuclides to specific cancer subtypes [5]. Xofigo® (Ra(²²³Ra)Cl₂), the first TAT drug approved by the US Food and Drug Administration, is a clear example of the limitations of targeting by chemical affinity because it can only be used to treat metastatic castration-resistant prostate cancer (mCRPC) that has spread to the bone [3]. Radioimmunoconjugates offer the possibility to treat different cancer subtypes by

using targeting vectors that will bind to antigens overexpressed on the surface of cancer cells [1,4,6]. Multiple $\alpha$-emitting radioimmunoconjugates are undergoing clinical trials as a treatment for different cancers, including mCRPC (phase 1, NCT03724747), solid tumors expressing mesothelin (phase 1, NCT03507452), and acute myeloid leukemia (phase 2, NCT03867682) [7–9]. These radioimmunoconjugates can readily deliver the $\alpha$-emitting radionuclide to the tumor site; however, they cannot prevent leakage of decay daughters from the tumor site or their potential relocation to healthy tissues and organs, due to their bond-breaking recoil energy (100–200 keV) [4,6]. Although the relocation of decay daughters can lead to unwanted radiotoxicity to healthy organs and a decrease in therapeutic efficacy, radioimmunoconjugates with rapid uptake into cancer cells can overcome these limitations and have shown remarkable therapeutic efficacy [10–12].

Organic and inorganic nanoparticles (NPs) have been proposed as delivery platforms for $\alpha$-emitting radionuclides to minimize the relocation of decay daughters from a tumor site. Organic NPs, including liposomes and polymersomes, prevent the relocation of radionuclides by containing all decay daughters within an aqueous volume enclosed by a hydrophobic bilayer composed of either phospholipids or block copolymers [13–16]. The retention of decay daughters within organic NPs is size-dependent, and the localization of $\alpha$-emitting radionuclides at the hydrophobic bilayer can lead to poor radionuclide retention [13–15]. Enhanced retention of decay daughters was obtained by precipitating $^{225}$Ac-InPO$_4$ nanocrystals within polymersomes due to the high stopping power (i.e., high-Z) environment surrounding the $\alpha$-emitting radionuclide [17]. The principle of surrounding $\alpha$-emitting radionuclides with a high stopping power environment has been used for developing inorganic NPs. Different compounds have been studied as delivery platforms to minimize the relocation of decay daughters, including lanthanide phosphate [18–21], lanthanide vanadate [22,23], iron oxide [24,25], titanium dioxide [26,27], zeolites [28,29], barium sulfate [30,31], and gold [32]. The retention of decay daughters within inorganic NPs is based on the stopping power of each compound, the NP density, and the fraction of recoil energy that can be transferred to the NP by translational, rotational, and vibrational modes [18,20,21,33]. Organic and inorganic NPs radiolabeled with $\alpha$-emitting radionuclides have been tested in vitro to assess the retention of radionuclides and their cytotoxicity against tumor cells and tumor spheroids [14,27,28,34]. In vivo evaluation of radiolabeled NPs has focused on the biodistribution of both NPs and radionuclides, their therapeutic efficacy against different cancers, and their side effects [21,25,32,35]. Both in vitro and in vivo experiments have shown the potential application of NPs as delivery platforms of $\alpha$-emitting radionuclides in clinical settings.

This work aims to develop lanthanum vanadate (LaVO$_4$) NPs as delivery platforms for $^{223}$Ra (t$_{1/2}$ = 11.4 d) and to enhance their radionuclide retention and surface chemistry for TAT. LaVO$_4$ NPs were studied as alternatives to other lanthanide-based NPs, GdVO$_4$, and LaPO$_4$, because LaVO$_4$ NPs have a simple synthesis and can be precipitated with monoclinic and tetragonal structures. The characteristic monoclinic structure of LaVO$_4$ NPs has exhibited higher radiation resistance and greater capacity to incorporate multivalent cations than the tetragonal structure of GdVO$_4$ NPs [36,37]. Additionally, LaVO$_4$ NPs can be synthesized by coprecipitation in aqueous media at room temperature in approximately 30 min [38], which can be exploited for radiolabeling with short-lived diagnostic radionuclides. The encapsulation and retention of radionuclides were optimized by adjusting the synthesis parameters of La($^{223}$Ra, $^{140}$Ba)VO$_4$ NPs, where $^{140}$Ba (t$_{1/2}$ = 12.8 d), a processing byproduct, was used to assess its potential as a surrogate of $^{223}$Ra. Retention studies were performed in vitro by dialyzing the NP suspension against deionized water or phosphate-buffered saline (PBS). Altering the surface chemistry of NPs with different compounds can enhance their colloidal stability and help them evade the reticuloendothelial system (RES), while providing functional groups for conjugation with targeting vectors and fluorophores [25,31,39,40]. To tackle this, we investigated the modification of LaVO$_4$ NP surface chemistry with different carboxylate and phosphate compounds.

## 2. Experimental Section

### 2.1. Materials and Reagents

The following reagents were purchased from Sigma-Aldrich (St. Louis, MO, USA) and were used without further purification: lanthanum(III) chloride heptahydrate (LaCl$_3$·7H$_2$O; ACS reagent, CAS 10025-84-0), ethylenediaminetetraacetic acid disodium salt dihydrate (EDTA; ACS/USP grade, CAS 6381-92-6), ammonium citrate dibasic (NH$_4$-Cit; ACS reagent, 98%, CAS 3012-65-5), sodium citrate tribasic dihydrate (Na-Cit; ACS reagent, ≥ 99%, CAS 6132-04-3), sodium phosphate crystals (Hex; +80 mesh, 96%, CAS 68915-31-1). Sodium orthovanadate (Na$_3$VO$_4$; 99%, Acros Organics, CAS 13721-39-6) and sodium tripolyphosphate (TPP; for analysis, Acros Organics, CAS 7758-29-4) were purchased from Fisher Scientific (Waltham, MA, USA) and used without further purification. Deionized water (18 Ω·cm) was obtained from a MilliporeSigma™ Milli-Q™ Ultrapure Water System. NP suspensions were dialyzed using 10K molecular weight cutoff Slide-A-Lyzer G2 Dialysis Cassettes (ThermoFisher, catalog number 87730).

Radionuclides $^{223}$Ra and $^{140}$Ba were obtained from the chemical-processing campaign designed to recover $^{225}$Ac from a proton-irradiated $^{232}$Th target. In that process, the $^{232}$Th target was initially dissolved in concentrated HCl, and the solution was loaded onto an anion exchange resin (BIO-RAD AG® MP-1M 100–200 mesh). Thorium(IV) was eluted from the column using 10 M HCl, and a matrix conversion to 1 M citric acid was performed before the Th eluate was loaded onto a cation exchange resin (BIO-RAD AG® 50W-X8 100–200 mesh). Subsequently, $^{223}$Ra and $^{140}$Ba were eluted from the column by using 2.5 M HNO$_3$, and the eluate was converted to 0.1 M HNO$_3$ for the synthesis of La($^{223}$Ra, $^{140}$Ba)VO$_4$ NPs. Carrier-free $^{223}$Ra was obtained from a $^{227}$Ac generator, where the parent radionuclide was produced via thermal neutron irradiation of a $^{226}$Ra target. The irradiated target was processed to chemically separate and purify $^{227}$Ac, which is dissolved in 0.1 M HNO$_3$ for further handling. The radioactive $^{227}$Ac solution was loaded into a cation exchange resin (BIO-RAD AG® 50W-X4 100–200 mesh), and then $^{223}$Ra was eluted with 8 M HNO$_3$. The $^{223}$Ra eluate was converted to 0.1 M HNO$_3$ for La($^{223}$Ra)VO$_4$ NP synthesis. The decay schemes of $^{223}$Ra and $^{140}$Ba are shown in Figure S1 in the electronic supplementary information (ESI).

### 2.2. Synthesis of La($^{223}$Ra, $^{140}$Ba)VO$_4$, La($^{223}$Ra)VO$_4$, and LaVO$_4$ Nanoparticles

A modified version of the procedure reported by Huignard et al. [38] was used to synthesize the La($^{223}$Ra, $^{140}$Ba)VO$_4$, La($^{223}$Ra)VO$_4$, and LaVO$_4$ NPs. La($^{223}$Ra, $^{140}$Ba)VO$_4$ and La($^{223}$Ra)VO$_4$ NPs were obtained by coprecipitation in aqueous media, where the radionuclides were mixed with a LaCl$_3$ solution, and then combined with a Na$_3$VO$_4$ solution at a 1:1 molar ratio following different steps (Figure 1). The proposed modifications were motivated by our previous work with GdVO$_4$ NPs [41]. For characterization purposes, nonradioactive LaVO$_4$ NPs were prepared following the same steps described for La($^{223}$Ra, $^{140}$Ba)VO$_4$ and La($^{223}$Ra)VO$_4$ NPs.

In procedure A, a 0.1 M HNO$_3$ solution containing 29 μCi of $^{223}$Ra, 21 μCi of $^{140}$Ba, and 18 μCi of $^{140}$La was evaporated to dryness in a Pyrex v-vial using an infrared heat lamp and a hot plate. A 0.5 mL LaCl$_3$ (0.1 M) solution was added into the Pyrex v-vial, stirred for 10 min, and transferred into a microcentrifuge tube. The LaCl$_3$/$^{223}$Ra/$^{140}$Ba solution was added dropwise to a 0.5 mL Na$_3$VO$_4$ (0.1 M, pH 12.5) solution under constant stirring, and the mixture was stirred for 30 min. Then, the La($^{223}$Ra, $^{140}$Ba)VO$_4$ NP suspension was transferred into the dialysis cassette to evaluate the encapsulation and retention of radionuclides over time.

In procedure B, a 0.1 M HNO$_3$ solution containing 15 μCi of $^{223}$Ra, 10 μCi of $^{140}$Ba, and 7 μCi of $^{140}$La was evaporated to dryness in a Pyrex v-vial using an infrared heat lamp and a hot plate, and the precipitate was mixed with a 0.5 mL LaCl$_3$ (0.1 M) solution for 10 min. A 0.5 mL Na$_3$VO$_4$ (0.1 M, pH 12.5) solution was added dropwise under constant stirring into the Pyrex v-vial containing the LaCl$_3$/$^{223}$Ra/$^{140}$Ba solution, and the mixture was stirred for 30 min before the NP suspension was transferred into the dialysis cassette.

In procedure C, a 0.1 M $HNO_3$ solution containing 20 µCi of $^{223}Ra$, 10 µCi of $^{140}Ba$, and 7 µCi of $^{140}La$ was evaporated to dryness in a Pyrex v-vial using an infrared heat lamp and a hot plate, and the precipitate was mixed with a 0.5 mL $LaCl_3$ (0.1 M) solution for 10 min. After dropwise addition of a 0.5 mL $Na_3VO_4$ (0.1 M, pH 12.5) solution into the $LaCl_3/^{223}Ra/^{140}Ba$ solution, the pH of the NP suspension was adjusted to 11 using KOH (1 M). The NP suspension was stirred for 30 min and then transferred into the dialysis cassette.

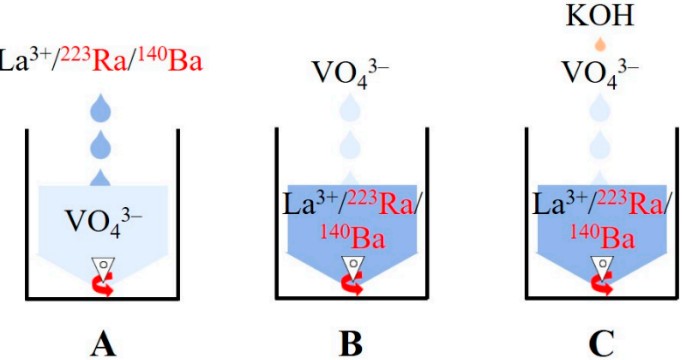

**Figure 1.** Schematic representation of the three synthesis procedures used to prepare $La(^{223}Ra, ^{140}Ba)VO_4$, $La(^{223}Ra)VO_4$, and $LaVO_4$ NPs.

## 2.3. Surface Modification of LaVO₄ Nanoparticles

$LaVO_4$ NPs, synthesized following procedure C, were modified with different compounds containing carboxylate (NH₄-Cit, Na-Cit, EDTA) or phosphate (TPP, Hex) groups to enhance their stability in aqueous media and to allow their functionalization with targeting vectors or fluorophores. Initially, each compound was distributed within a microcentrifuge tube to yield different $LaVO_4$:compound molar ratios (e.g., 6:1, 3:1, 1:1, and 1:2), based on a $LaVO_4$ NP concentration of 10.9 mg/mL. The as-prepared $LaVO_4$ NP suspension was transferred into the microcentrifuge tube containing the respective compound and was mixed vigorously using a vortex mixer for 30 min. The modified $LaVO_4$ NPs were washed three times with deionized water (14,500 rpm, 20 min) to remove unreacted species.

## 2.4. Characterization of LaVO₄ Nanoparticles

The $LaVO_4$ NP precipitate was suspended in aqueous media or evaporated to dryness in a muffle furnace at 60 °C for 6 to 8 h. The NP suspensions were diluted 50 times in deionized water or PBS (1X, pH 7.2). Dynamic light scattering and phase analysis light scattering were performed on a ZetaPALS zeta potential analyzer (Brookhaven Instruments Corporation, Holtsville, NY, USA) to characterize the particle size distribution and the surface charge. The dried precipitate was ground using a mortar and pestle to obtain a powder sample that was used for X-ray diffraction, Fourier-transform infrared (FTIR), and morphological characterization. Powder X-ray diffraction was performed on an AXRD® Benchtop diffractometer (PROTO Manufacturing Inc., Taylor, MI, USA) operated at 30 kV and 20 mA with a 1500 W fine-focus Cu ceramic X-ray tube (λ = 1.5406 Å). Rietveld refinement was used to determine the crystal structure of $LaVO_4$ NPs. Surface characterization was carried out using attenuated total reflectance FTIR spectroscopy on a Nicolet™ iS50 FTIR spectrometer with a diamond crystal (Thermo Scientific™, Waltham, MA, USA). A Hitachi 300 kV cold-field emission gun transmission electron microscope was used to characterize the morphology of NP powders that had been dispersed in isopropanol and drop-casted onto 200 mesh Lacey carbon copper grids.

## 2.5. Radionuclide Encapsulation within La(²²³Ra,¹⁴⁰Ba)VO₄ and La(²²³Ra)VO₄ Nanoparticles

$La(^{223}Ra, ^{140}Ba)VO_4$ NPs were transferred into a dialysis cassette after synthesis and were dialyzed against deionized water overnight to remove unreacted species and radionu-

clides. A 5 mL dialysate aliquot was removed before the dialysate was replaced with deionized water or PBS (1X, pH 7.4) to assess the radionuclide encapsulation over time. Dialysis of La($^{223}$Ra)VO$_4$ NPs, synthesized following procedure C, against PBS was performed to expose the NP suspensions to biologically relevant conditions. A 5 mL dialysate aliquot was taken periodically, every 2 to 3 days, and was assayed by γ-ray spectroscopy to determine the activity of $^{223}$Ra, $^{211}$Pb, $^{211}$Bi, $^{140}$Ba, and $^{140}$La within the dialysate. A high-purity germanium detector (Ortec, Oak Ridge, TN, USA), having a ~100 cm$^3$ crystal active volume and a beryllium window, coupled to a PC-based multichannel analyzer (Mirion Technologies, San Ramon, CA, USA), was used for γ-ray spectroscopy. Energy and efficiency calibrations were determined by γ-ray sources traceable to the National Institute of Standards and Technology. The γ-ray energies and intensities used to calculate the activity of each radionuclide are summarized in Table S1 in the ESI. Equation (1) was used to calculate the fraction of activity found in the dialysate, referred to as leakage:

$$Leakage(\%) = \frac{A_{da} \times V_c}{A_{DC}(t)} \times 100 \qquad (1)$$

where $A_{da}$ is the activity measured in the dialysate aliquot, $V_c$ is a volume correction to account for the total activity in the dialysate (i.e., $V_c = V_{\text{dialysate}} / V_{\text{dialysate aliquot}}$), and $A_{DC}(t)$ is the expected activity within the dialysis cassette corrected by radioactive decay. The $^{223}$Ra, $^{140}$Ba, and $^{140}$La radiochemical yields were calculated based on the initial activity and the activity lost during synthesis and dialysis as summarized in Equation (2):

$$RY(\%) = \frac{A_0 - (A_{VS} + A_D + A_{TP} + A_{D_{max}})}{A_0} \times 100 \qquad (2)$$

where $A_0$ is the initial activity, $A_{VS}$ is the activity lost in the vial and stirrer, $A_D$ is the activity lost in the first dialysis, $A_{TP}$ is the activity lost in the transfer pipette and microcentrifuge tube, and $A_{D_{max}}$ is the maximum activity lost over time during dialysis. All activities were corrected for radioactive decay. To assess the influence of surface modification on the $^{223}$Ra radiochemical yield, La($^{223}$Ra)VO$_4$ NP suspensions were mixed at a 1:1 (NH$_4$–Cit, Na–Cit, Hex, or TPP) and 3:1 (EDTA) LaVO$_4$:compound molar ratios. Modified La($^{223}$Ra)VO$_4$ NPs were washed three times with deionized water (14,500 rpm, 20 min), and each supernatant was carefully removed with a transfer pipette and stored in a microcentrifuge tube for characterization using γ-ray spectroscopy. La($^{223}$Ra)VO$_4$ NP suspensions were dispersed in deionized water and characterized using γ-ray spectroscopy to determine the activity of $^{223}$Ra after surface modification and cleaning.

## 3. Results and Discussion

### 3.1. Characterization of LaVO$_4$ NPs

The procedures used to synthesize LaVO$_4$ NPs were variations on a procedure developed for YVO$_4$:Eu NPs [38], where the order in which LaCl$_3$ and Na$_3$VO$_4$ solutions are mixed was varied (Figure 1). The modified procedures rely on the precipitation of La$^{3+}$ and VO$_4^{3-}$ ions in aqueous media to form LaVO$_4$ NPs [38]. Procedure A is similar to the one developed by Huignard et al. for YVO$_4$:Eu NPs, with the difference that the pH of the NP suspension was not adjusted to 11 after dropwise addition of LaCl$_3$ into Na$_3$VO$_4$. Since no additional reagents were added during synthesis, the pH of the LaVO$_4$ NP suspensions was 4.9. A similar pH was obtained for LaVO$_4$ NP suspensions prepared following procedure B, where the Na$_3$VO$_4$ solution was added dropwise into the LaCl$_3$ solution. In procedure C, both LaCl$_3$ and Na$_3$VO$_4$ solutions were mixed in a similar fashion as carried out in procedure B, with the difference that KOH was used to adjust the pH of the LaVO$_4$ NP suspension to 11. It is expected that the order in which both LaCl$_3$ and Na$_3$VO$_4$ are mixed can influence the encapsulation of surrogate cations and radionuclides within LaVO$_4$ NPs [41]. A low pH can lead to the precipitation of vanadate species that are different from VO$_4^{3-}$, whereas a pH >13 will lead to the precipitation of hydroxide species,

such as La(OH)$_3$ [38,42,43]. Overall, this synthesis procedure is attractive for radiochemical settings, given its simplicity and the fact that different dopant cations can be incorporated into LaVO$_4$ NPs to provide luminescent and/or magnetic properties.

The diffraction patterns of as-prepared LaVO$_4$ NPs, synthesized by the different procedures, match the reference patterns of both tetragonal (powder diffraction file: 00-032-0504, space group I4$_1$/amd) and monoclinic (powder diffraction file: 00-050-0367, space group P2$_1$/n) crystal structures (Figure 2). Both tetragonal and monoclinic structures have been reported for LaVO$_4$ NPs [44,45]. Since the monoclinic structure is thermodynamically stable, due to its high coordination number (CN; CN = 9) and the large ionic radius of La$^{3+}$ (1.216 Å for CN = 9 [46]), it is expected that LaVO$_4$ NPs will precipitate primarily with a monoclinic structure [47,48]. Although the tetragonal structure is preferred for smaller lanthanides (i.e., Ln = Ce to Lu), the precipitation of LaVO$_4$ having a tetragonal structure can be obtained by tailoring the synthesis method, the pH of the reaction, and the dopant concentration [49–52]. LaVO$_4$ NPs, synthesized by procedures A and B, exhibited Bragg peaks with similar intensities, whereas additional Bragg peaks associated with the monoclinic structure were observed for LaVO$_4$ NPs prepared by procedure C (Figure 2). The fraction of each crystal structure and its crystallite size (Table 1) were determined by Rietveld refinement and the Scherrer equation, respectively. The crystallite size was calculated from the (200) plane (i.e., 2 Theta = 23°) in the tetragonal structure and the (012) plane (i.e., 2 Theta = 30°) in the monoclinic structure. Based on Rietveld refinement, a large fraction of LaVO$_4$ NPs prepared using procedure A were characterized by a tetragonal structure (66.1%). The precipitation of LaVO$_4$ NPs with a tetragonal structure may have been influenced by the low pH (4.9) at which the reaction was kept for 30 min. LaVO$_4$ NPs prepared using procedure B were characterized by a similar, yet a distinct fraction of both tetragonal and monoclinic structures (Table 1), suggesting that the order in which the reagents were mixed influences the crystal structure of LaVO$_4$ NPs. In procedure C, Rietveld refinement revealed that a large fraction of LaVO$_4$ NPs was characterized by a monoclinic structure (Table 1), suggesting that adjusting the pH to 11 promoted the precipitation of particles with a thermodynamically stable structure. The adjustment of the suspension's pH can lead to changes in the interfacial tension, and hence, differences in nucleation and growth kinetics. The changes in nucleation and growth kinetics can contribute to the precipitation of particles with different crystal structure fractions [49].

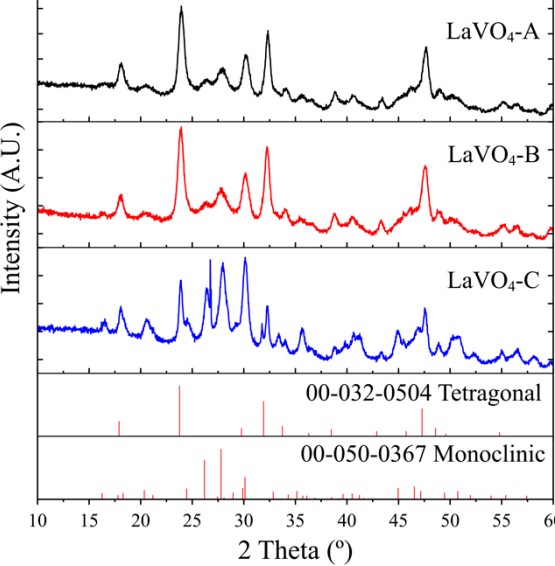

**Figure 2.** The crystal structure of LaVO$_4$ NPs was influenced by the synthesis procedure and the suspension's pH. Diffraction patterns of LaVO$_4$ NPs, synthesized following different procedures (A, B, and C), are compared with reference powder diffraction files for a LaVO$_4$ compound having tetragonal (00-032-0504) and monoclinic (00-050-0367) crystal structures.

**Table 1.** The tetragonal and monoclinic structure fraction and crystallite size of LaVO$_4$ NPs, synthesized by different procedures. The fraction of crystal structure was influenced by the synthesis procedure and the suspension's pH.

| Procedure | Tetragonal | | Monoclinic | |
|:---:|:---:|:---:|:---:|:---:|
| | Fraction (%) | Crystallite Size (nm) | Fraction (%) | Crystallite Size (nm) |
| A | 66.1 | 34.6 | 33.9 | 36.9 |
| B | 46.4 | 45.5 | 53.6 | 32.3 |
| C | 17.6 | 61.9 | 82.4 | 38.3 |

Transmission electron micrographs showed that the precipitation of LaVO$_4$ NPs had different morphologies, depending on the synthesis procedure followed (Figure 3). The aggregation of LaVO$_4$ NPs, which was a common characteristic between the different procedures, could be promoted by the sample preparation, the lack of stabilizing species on the particle surface, or both. LaVO$_4$ NPs prepared by procedure A were represented by particles having an anisotropic and elongated morphology, analogous to a tetragonal shape (Figure 3a,d). In Figure 3d, the lattice fringes of LaVO$_4$ NPs correspond to an interplanar spacing of 3.2 Å and 3.6 Å associated with the (120) and (200) planes of the monoclinic and tetragonal structures, respectively. Particles having a more defined tetragonal shape were observed in LaVO$_4$ NPs, synthesized following procedure B (Figure 3b,e). The interplanar spacing between lattice fringes corresponds to both monoclinic (d$_{120}$ = 3.2 Å) and tetragonal (d$_{200}$ = 3.6 Å) structures (Figure 3e). LaVO$_4$ NPs prepared using procedure C were characterized by a small fraction of large particles having a tetragonal shape and a large fraction of aggregated anisotropic particles (Figure 3c,f). These results are consistent with the characteristic morphology obtained for GdVO$_4$ NPs when the pH was adjusted to >11, where both large trapezoidal particles and aggregates of small crystallites were observed [41]. Based on an interplanar spacing of 3.5 Å and 3.3 Å, the lattice fringes correspond to the (120) and (200) planes of the monoclinic structure (Figure 3f).

The surface chemistry of LaVO$_4$ NPs, synthesized by the different procedures, was characterized by similar vibration bands (Figure 4). The La–O bond is represented by a weak vibration band at 433 cm$^{-1}$, whereas the strong vibration at 765 cm$^{-1}$ corresponds to the V–O bond [47,48]. The bands centered at 1397 cm$^{-1}$ and 1455 cm$^{-1}$ are related to the symmetrical and asymmetrical vibrations of carboxylate groups, respectively [45,53]. The presence of carboxylate groups can be associated with the formation and precipitation of carbonate species during synthesis, promoted by a high reactivity between the surface hydration layers of LaVO$_4$ NPs and the CO$_2$ from the air [45,53]. The bands located at 1634 cm$^{-1}$ and 3300 cm$^{-1}$ correspond to the bending and stretching O–H vibrations from physically absorbed water molecules on the surface of LaVO$_4$ NPs [43,54].

As shown by the transmission electron micrographs, LaVO$_4$ NPs, synthesized by different procedures exhibited significant aggregation (Figure 3). These results were supported by the average hydrodynamic diameter obtained from the intensity distribution after the characterization of LaVO$_4$ NP suspensions using dynamic light scattering. LaVO$_4$ NPs prepared by different procedures had a hydrodynamic diameter >900 nm when diluted in deionized water or PBS (Table 2). Particle aggregation can be associated with hydrogen bonding between surface hydroxyl groups [40]. There was a significant difference between the zeta potential of LaVO$_4$ NPs diluted in deionized water or PBS, depending on the synthesis procedure followed (Table 2). When LaVO$_4$ NPs were diluted in deionized water, LaVO$_4$ NPs, synthesized following procedures A and B, had a positive zeta potential (~17 mV), whereas LaVO$_4$ NPs prepared by procedure C were characterized by a negative zeta potential (−26.5 mV). This difference in zeta potential could be attributed to a higher fraction of carbonate species on the surface of LaVO$_4$ NPs prepared by procedure C, as shown from the FTIR spectrum (Figure 4). Dilution of LaVO$_4$ NPs in PBS promoted a significant change in the magnitude of the zeta potential that depends on the synthesis procedure followed (Table 2). The change in zeta potential could be related to the adsorption or

interaction of phosphate species with the LaVO$_4$ NP surface, which can be concluded based on the vibration bands of phosphate groups observed on the FTIR spectrum (Figure S2 in the ESI). It is expected that phosphate species from PBS will have a strong interaction with OH groups and La cations on the particle surface.

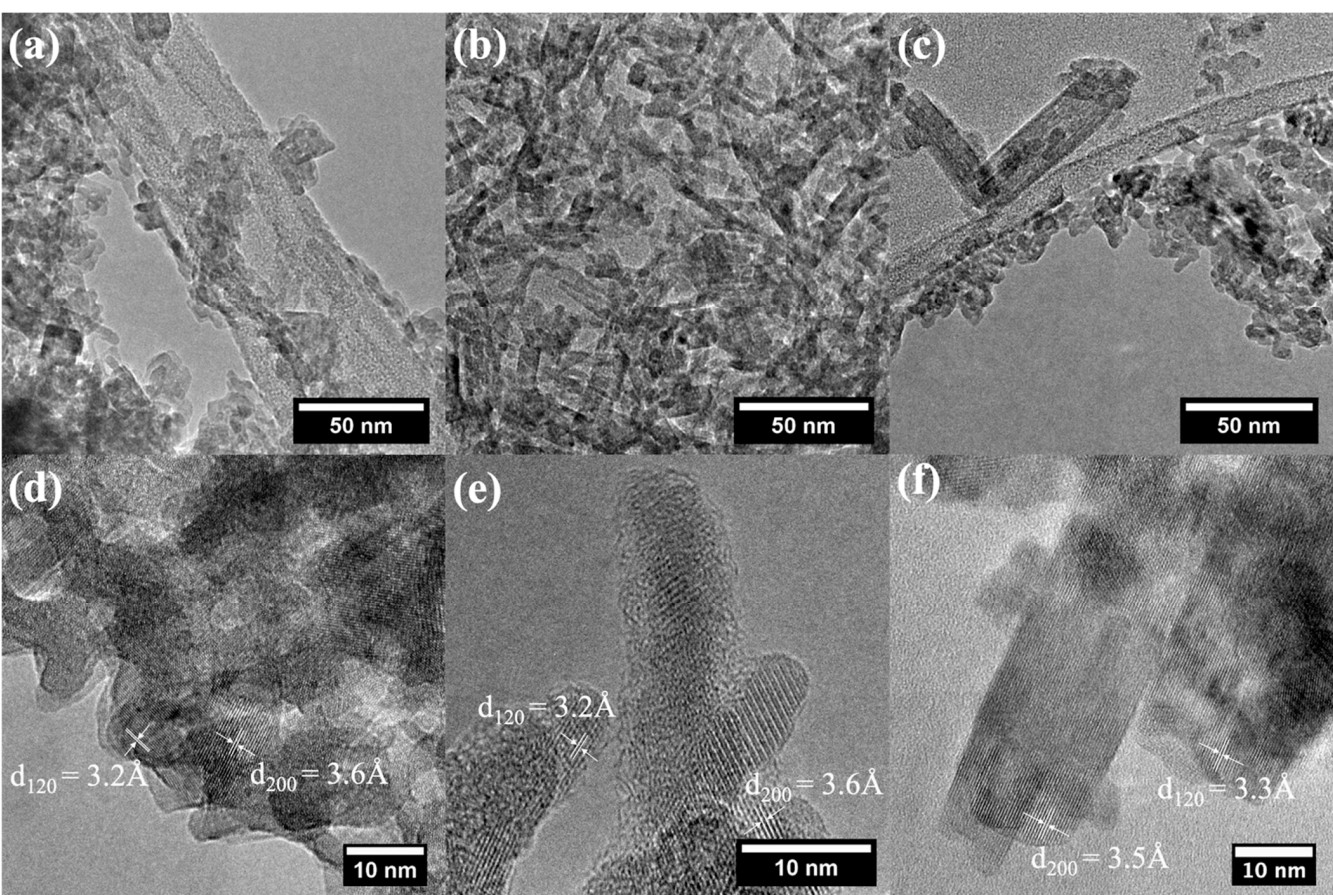

**Figure 3.** The synthesis procedure influenced the morphology of the LaVO$_4$ NPs. Transmission electron micrographs of LaVO$_4$ NPs, synthesized using procedure A (**a,d**), B (**b,e**), and C (**c,f**).

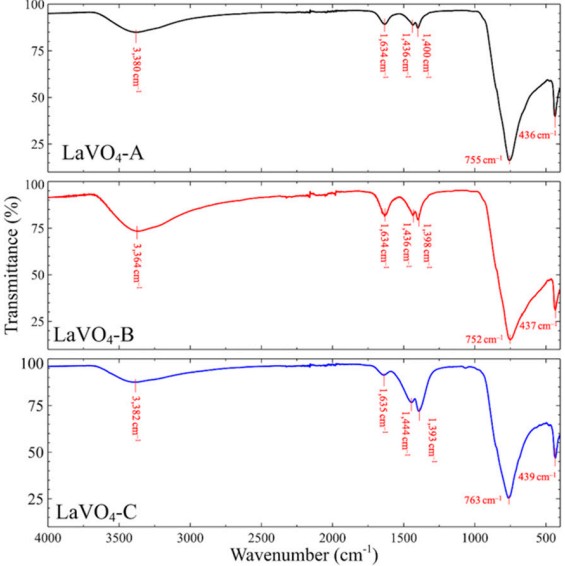

**Figure 4.** FTIR spectra of the LaVO$_4$ NPs, synthesized by procedures A, B, and C, show no significant difference in their surface chemistry.

**Table 2.** Intensity diameter and zeta potential of diluted LaVO$_4$ NPs in deionized water and PBS.

| Procedure | Dilution in Deionized Water | | Dilution in PBS | |
|---|---|---|---|---|
| | Diameter (nm) | Zeta Potential (mV) | Diameter (nm) | Zeta Potential (mV) |
| A | $1186 \pm 129$ | $17.4 \pm 1.8$ | $1036 \pm 101$ | $28.1 \pm 7.9$ |
| B | $1085 \pm 97$ | $17.6 \pm 3.2$ | $973 \pm 12$ | $-19.2 \pm 4.6$ |
| C | $995 \pm 93$ | $-26.5 \pm 8.4$ | * | $-3.9 \pm 3.4$ |

* Particle size distribution was not measured, due to the poor stability of LaVO$_4$ NP suspension.

### 3.2. Surface-Modified LaVO$_4$ NPs

The nature of the functional groups of LaVO$_4$ NPs synthesized by the different procedures resulted in poor colloidal stability and limits their functionalization with targeting vectors. To address that limitation, LaVO$_4$ NPs prepared following procedure C were modified through surface adsorption and/or complexation with different compounds containing carboxylate and phosphate groups. Additional vibrational bands were observed after LaVO$_4$ NPs were mixed with each compound at room temperature for 30 min (Figure 5 and Figures S3–S7 in the ESI). The characteristic vibrational bands of the LaVO$_4$ NPs modified with the different compounds are summarized in Table 3. The FTIR spectra of the LaVO$_4$ NPs modified with NH$_4$-Cit showed that at least a 3:1 LaVO$_4$:NH$_4$-Cit molar ratio is required to obtain a significant difference in the transmittance of the vibrational bands and the appearance of the asymmetric carboxylate bidentate bond stretching band at 1557 cm$^{-1}$ (Figure S3 in the ESI). Additionally, the total mass of LaVO$_4$ NPs was reduced by 12% after surface functionalization with a 1:1 LaVO$_4$:NH$_4$-Cit, suggesting a partial dissolution of the particles.

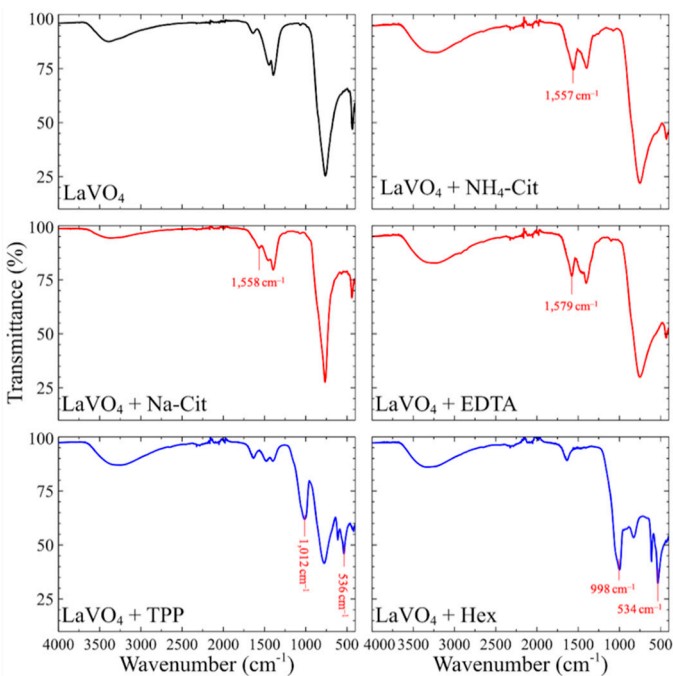

**Figure 5.** Surface modification of LaVO$_4$ NPs with stabilizing compounds increases the presence of surface functional groups, as shown by the detection of additional vibrational bands. FTIR spectra of LaVO$_4$ NPs, synthesized by procedure C (upper left), and after surface modification with NH$_4$-Cit, Na-Cit, Hex, or TPP (1:1 LaVO$_4$:compound molar ratio) and EDTA (3:1 LaVO$_4$:EDTA molar ratio).

Alternatively, when using Na-Cit to modify LaVO$_4$ NPs, the asymmetric carboxylate bidentate bond stretching band at 1557 cm$^{-1}$ was only observed at a 1:1 LaVO$_4$:Na-Cit molar ratio (Figure S4 in the ESI). The FTIR spectra from LaVO$_4$ NPs modified with Na-Cit showed the presence of carbonate species at different LaVO$_4$:Na-Cit molar ratios, based

on the symmetric and asymmetric vibrations of carboxylate groups [45,53]. A change of <4% in the mass of the $LaVO_4$ NPs after surface modification with 1:1 $LaVO_4$:Na-Cit molar ratio, suggests that Na-Cit can interact with La cations on the particle surface without causing a significant loss of mass. Modifying $LaVO_4$ NPs with EDTA resulted in the appearance of the asymmetric carboxylate bidentate bond stretching band and its shift to 1579 cm$^{-1}$ at both 6:1 and 3:1 $LaVO_4$:EDTA molar ratios (Figure S5 in the ESI). The shift of the carboxylate bidentate bond stretching band of $LaVO_4$ + EDTA with respect to that of $LaVO_4$ + $NH_4$-Cit and $LaVO_4$ + Na-Cit is likely associated with the coordination of the carboxylate groups around the La cation [55]. At a 3:1 $LaVO_4$:EDTA molar ratio, $LaVO_4$ + EDTA NPs were partially dissolved based on a 44% decrease in $LaVO_4$ NP mass during surface modification. It is assumed that higher concentrations of EDTA (i.e., 1:1 and 1:2 $LaVO_4$:EDTA molar ratios) completely dissolved $LaVO_4$ NPs because of a cation exchange reaction between $LaVO_4$ NPs and EDTA, leading to the formation of La-EDTA complexes [56].

**Table 3.** Characteristic vibrations and group assignment of $LaVO_4$ NPs, synthesized by different procedures and modified with different stabilizing compounds.

| Wavenumber Range (cm$^{-1}$) | Group Assignment |
|---|---|
| 425.3–441.8 | La–O bond |
| 533.4–542.8 | $PO_4$ bending vibration |
| 611.0–612.0 | $PO_4$ bending vibration |
| 743.4–825.8 | V–O vibration |
| 998.2–1032.0 | $PO_4$ stretching vibration |
| 1389.5–1407.7 | Symmetric vibration carboxylate groups (carbonate species) |
| 1434.2–1485.7 | Asymmetric vibration carboxylate groups (carbonate species) |
| 1553.2–1579.8 | Asymmetric carboxylate bidentate bond stretching |
| 1632.5–1638.6 | O–H bending |
| 3208.2–3382.2 | O–H stretching |

Modification of $LaVO_4$ NPs with phosphate-containing compounds (TPP and Hex), resulted in the appearance of characteristic phosphate bending and stretching vibration bands at all molar ratios (Figures S6 and S7 in the ESI). Bending vibration bands from $PO_4$ groups are observed at 536 cm$^{-1}$ and 611 cm$^{-1}$, whereas a strong stretching vibration band from $PO_4$ groups appears at 1013 cm$^{-1}$ [57,58]. The fact that the characteristic vibration bands from $PO_4$ groups are displayed after modification with a molar ratio of 6:1 $LaVO_4$:TPP and $LaVO_4$:Hex suggests a strong interaction between the phosphate groups and the surface La cations. Phosphate compounds can exhibit enhanced selectivity toward La cations relative to carboxylic compounds, due to their tetrahedral phosphonic group and the polarizability of the P–O bond [59]. Due to the strong interaction between La cations and phosphate compounds [60], $LaVO_4$ NPs were partially dissolved after surface modification with TPP and Hex (1:1 $LaVO_4$:compound molar ratio), as shown by a decrease in mass of 15% and 55%, respectively. Using Hex as a stabilizing compound may provide La cations with a hydrophobic environment to prevent their transmetalation when testing them in vivo [61].

Surface modification with the different compounds enhanced the colloidal stability of $LaVO_4$ NPs in deionized water based on their particle size distribution and zeta potential. The different compounds yielded a hydrodynamic diameter <300 nm (intensity distribution) and a negative zeta potential (Table 4). $LaVO_4$ + $NH_4$-Cit NPs had the smallest hydrodynamic diameter (144.7 ± 4.8 nm) and lowest zeta potential (−17.0 ± 1.8 mV) among the different compounds tested in deionized water. Although the zeta potential of $LaVO_4$ NPs prepared by procedure C before and after surface modification was not significantly different, a threefold decrease in the hydrodynamic diameter was obtained after surface modification. Surface-modified $LaVO_4$ NPs exhibited significant aggregation when diluted in PBS, resulting in a hydrodynamic diameter >1000 nm. The aggregation of $LaVO_4$ NPs in PBS can be attributed to a compression of the electric double layer induced



by the high ionic strength of PBS (1X, 162.7 mM [62]), and thus, a decrease in the magnitude of the zeta potential (Table 4) [40]. The desorption of carboxylate compounds through an exchange with phosphate species in PBS could also cause the aggregation of surface-modified $LaVO_4$ NPs based on the presence of characteristic $PO_4$ vibration bands on the FTIR spectra (Figure S8 in the ESI) [40]. These results suggest a weak interaction between carboxylate compounds with surface La cations, which could be detrimental for biological applications where proteins, such as albumin, can bind to La cations via a ligand exchange reaction [63]. The binding of proteins can affect the biodistribution and pharmacokinetics of NPs and lead to NPs being taken up by the RES [64].

**Table 4.** The colloidal stability of $LaVO_4$ NPs, in deionized water, is enhanced after surface modification. Hydrodynamic diameter (intensity distribution) and zeta potential of $LaVO_4$ NPs, synthesized by procedure C after surface modification with $NH_4$-Cit, Na-Cit, Hex, or TPP (1:1 $LaVO_4$:compound molar ratio) and EDTA (3:1 $LaVO_4$:EDTA molar ratio).

| Stabilizing Compound | Dilution in Deionized Water | | Dilution in PBS | |
|---|---|---|---|---|
| | Diameter (nm) | Zeta Potential (mV) | Diameter (nm) | Zeta Potential (mV) |
| None | $995 \pm 93$ | $-26.5 \pm 8.4$ | * | $-3.9 \pm 3.4$ |
| $NH_4$-Cit | $144.7 \pm 4.8$ | $-17.0 \pm 1.8$ | $1003 \pm 65$ | $-12.4 \pm 3.5$ |
| Na-Cit | $207.1 \pm 27.5$ | $-27.1 \pm 3.6$ | $1110 \pm 144$ | $-21.4 \pm 4.0$ |
| EDTA | $267.8 \pm 11.8$ | $-32.9 \pm 2.4$ | $1126 \pm 182$ | $-8.4 \pm 2.1$ |
| TPP | $209.1 \pm 5.2$ | $-30.5 \pm 2.1$ | $1112 \pm 157$ | $-15.8 \pm 2.1$ |
| Hex | $231.4 \pm 14.4$ | $-33.6 \pm 1.6$ | $1100 \pm 40$ | $-21.4 \pm 4.0$ |

* Particle size distribution was not measured, due to the poor stability of $LaVO_4$ NP suspension.

### 3.3. Encapsulation of $^{223}$Ra and Decay Daughters in La($^{223}$Ra)VO$_4$ NPs

The encapsulation of $^{223}$Ra and retention of its decay daughters ($^{211}$Pb and $^{211}$Bi) were assessed in vitro by dialyzing the La($^{223}$Ra)VO$_4$ NP suspensions against deionized water [19,22,23]. The leakage of radionuclides was calculated using Equation (1) and evaluated over time (Figure 6). La($^{223}$Ra, $^{140}$Ba)VO$_4$ NPs prepared following procedure A exhibited the highest leakage of $^{223}$Ra, $^{140}$Ba, and their decay daughters (Figure 6a). The leakage of $^{223}$Ra reached a maximum of $25.3 \pm 3.9\%$ and then decreased to $3.9 \pm 1.3\%$ after 10 days in dialysis (Figure 6a). A similar trend was observed for $^{140}$Ba, where its leakage was ~15% for the first two days and subsequently decreased to ~2.5% (Figure 6a). The high leakage observed for La($^{223}$Ra, $^{140}$Ba)VO$_4$ NPs prepared using procedure A can be attributed to the rapid precipitation of La, $^{223}$Ra, and $^{140}$Ba as hydroxides, due to the high pH of the Na$_3$VO$_4$ solution (pH = 12.5) [38]. Particularly, the precipitation of hydroxide species has been associated with a slow interaction with vanadate ions that can lead to a low yield of $LaVO_4$ NPs [38], which is shown by the low radiochemical yield obtained for both $^{223}$Ra ($19.2 \pm 4.1\%$) and $^{140}$Ba ($16.5 \pm 2.0\%$) (Figure 7). It is also expected that the formation of different radionuclide species due to pH changes will promote the uptake of radionuclides by surface sorption [65]. The sorption of radionuclides onto the particle surface could explain the high leakage observed for both $^{223}$Ra and $^{140}$Ba (Figure 6a). Differences in the radiochemical yield of divalent radionuclides ($^{223}$Ra, $^{140}$Ba) with respect to trivalent $^{140}$La can be associated with their mechanism of incorporation within the crystal structure [66]. For both $^{223}$Ra and $^{140}$Ba, it will be harder to incorporate into tetragonal $LaVO_4$ NPs and occupy La$^{3+}$ sites, due to the smaller La–O length with respect to monoclinic $LaVO_4$ NPs (2.5089 Å vs. 2.6029 Å) [52].

Among all the NPs produced by the different procedures, La($^{223}$Ra, $^{140}$Ba)VO$_4$ NPs prepared using procedure B exhibited the lowest leakage of radionuclides (Figure 6b). Although the radiochemical yield of $^{223}$Ra and $^{140}$Ba increased with respect to those obtained by procedure A, NPs prepared following procedure B had a relatively low radiochemical yield for the different radionuclides (Figure 7). It is hypothesized that precipitation of vanadate species different from $LaVO_4$, due to the low pH of the NP suspension [38], contributed to the low radiochemical yields. The precipitation of other kinds of lanthanum

vanadate species was prevented in procedure C by the addition of KOH to adjust the pH of the NP suspension to 11. The radionuclide leakage increased slightly with respect to La($^{223}$Ra, $^{140}$Ba)VO$_4$ NPs prepared using procedure B (Figure 6c), where the leakage of $^{223}$Ra and $^{140}$Ba reached a maximum of 4.6 ± 1.5% and 2.0 ± 1.8%, respectively. It is assumed that the dialysate activity of $^{223}$Ra and $^{140}$Ba may originate from radionuclides that were not coprecipitated within LaVO$_4$ NPs, but rather adsorbed onto the particle surface [65]. The leakage of decay daughters $^{211}$Pb and $^{211}$Bi was maintained at less than 1.5% over time (Figure 6c), while the $^{223}$Ra, $^{140}$Ba, and $^{140}$La radiochemical yield increased to 87.4 ± 6.9%, 89.0 ± 8.3%, and 94.2 ± 4.3%, respectively. Overall, La($^{223}$Ra, $^{140}$Ba)VO$_4$ NPs prepared using procedure C exhibited an enhanced radiochemical yield for both radium and barium radionuclides compared with that obtained with BaSO$_4$ NPs (20% for >100 nm and 31–40% for <10 nm) [30,31].

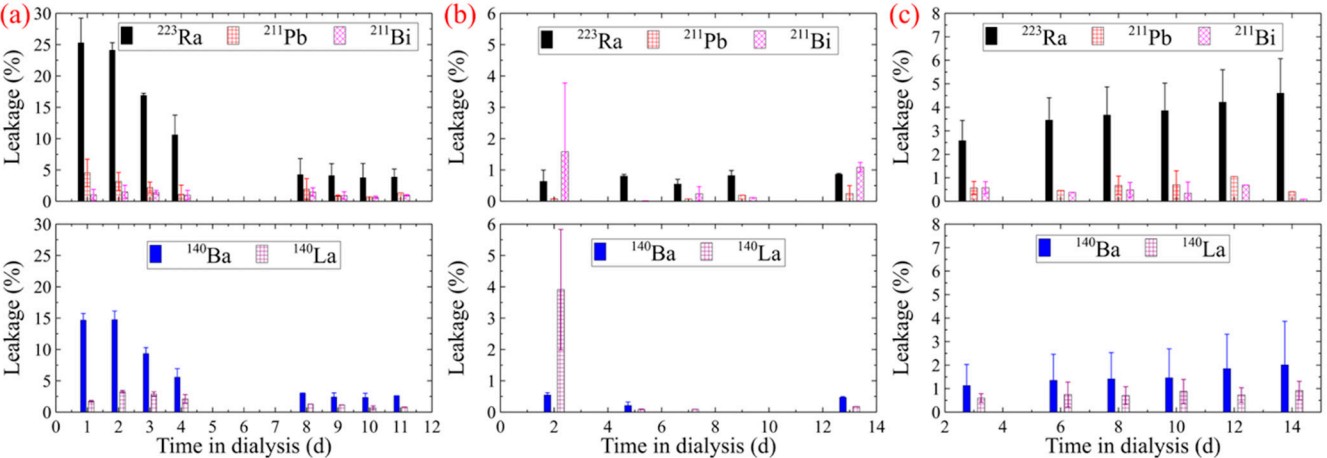

**Figure 6.** Radionuclide leakage of La($^{223}$Ra, $^{140}$Ba)VO$_4$ NPs, synthesized by procedure (**a**) A, (**b**) B, and (**c**) C (*n* = 3), over time. La($^{223}$Ra, $^{140}$Ba)VO$_4$ NPs were dialyzed against deionized water, and dialysate aliquots were taken periodically and assayed using γ-ray spectroscopy.

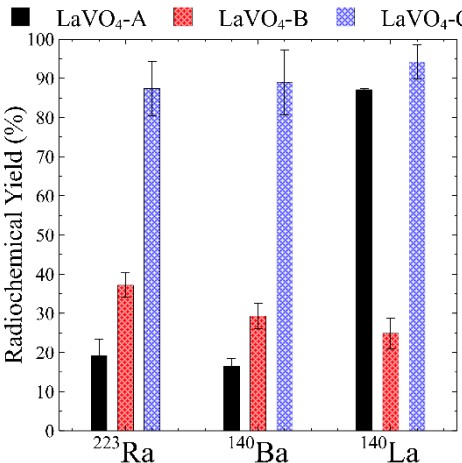

**Figure 7.** The radiochemical yield of La($^{223}$Ra, $^{140}$Ba)VO$_4$ NPs, synthesized by different procedures (*n* = 3). Enhanced radiochemical yield was observed for La($^{223}$Ra, $^{140}$Ba)VO$_4$ NPs, synthesized using procedure C.

The $^{223}$Ra radiochemical yield for La($^{223}$Ra, $^{140}$Ba)VO$_4$ NPs was similar to that obtained with La($^{223}$Ra)PO$_4$ core NPs (91%) and higher than that for core + 2 shells (80%) [19]. Although the leakage of radionuclides was <5% for both BaSO$_4$ and La($^{223}$Ra, $^{140}$Ba)VO$_4$ NPs, La($^{223}$Ra)PO$_4$ core + 2 shells NPs exhibited a greater retention of decay daughters (<0.1% leakage) [19,30,31]. The enhanced retention of decay daughters can be associated

with the core-shell structure of those NPs, where the nonradioactive shell acts as a barrier to prevent the relocation of radionuclides [18–20,22,23].

The stability and radionuclide retention of La($^{223}$Ra)VO$_4$ NPs, synthesized by procedure, C were assessed by dialyzing the NP suspension against PBS (Figure 8). The $^{223}$Ra radiochemical yield calculated after dialysis against PBS for 15 days was 99.7 ± 0.2% (*n* = 3), which is comparable with that reported for intrinsically radiolabeled hydroxyapatite (Hap; 97.0 ± 0.5%) and titanium dioxide (TiO$_2$; 99.1 ± 0.3%) NPs [26]. Although the leakage of decay daughters was not reported, the leakage or released activity of $^{223}$Ra from Hap NPs was significantly higher with respect to that of La($^{223}$Ra)VO$_4$ NPs [26]. The decrease in radionuclide leakage after dialysis of La($^{223}$Ra)VO$_4$ NPs against PBS may be related to the interaction of free radionuclides with phosphate species in solution, changes in NP colloidal stability, or both (Figure 8). It is hypothesized that the sorption and surface complexation of $^{223}$Ra and phosphate compounds in solution with LaVO$_4$ NPs decreases the radionuclide activity in the dialysate [24,65]. The interaction of phosphate species with LaVO$_4$ NPs after dilution in PBS was shown in their FTIR spectra (Figure S2 in the ESI). To address the influence of surface modification in the $^{223}$Ra radiochemical yield, La($^{223}$Ra)VO$_4$ NPs were modified with carboxylate and phosphate compounds, centrifuged, and dispersed in deionized water multiple times. Surface modification with Na-Cit did not significantly alter the $^{223}$Ra radiochemical yield (99.6 ± 2.3%) with respect to La($^{223}$Ra)VO$_4$ NPs (Figure 9). The $^{223}$Ra radiochemical yield of La($^{223}$Ra)VO$_4$ NPs modified with NH$_4$-Cit, TPP, and EDTA decreased ~15%, whereas >25% of $^{223}$Ra initial activity was lost when using Hex as a stabilizing compound (Figure 9). The decrease in $^{223}$Ra radiochemical yield after surface modification is related to the interaction of those stabilizing compounds with La cations on the particle surface and their ability to form La complexes. As shown with LaVO$_4$ NPs, there is a mass loss associated with each stabilizing compound used for surface functionalization, which could lead to a decrease in $^{223}$Ra radiochemical yield. Also, the selectivity of stabilizing compounds to specific cations can lead to a lower radiochemical yield.

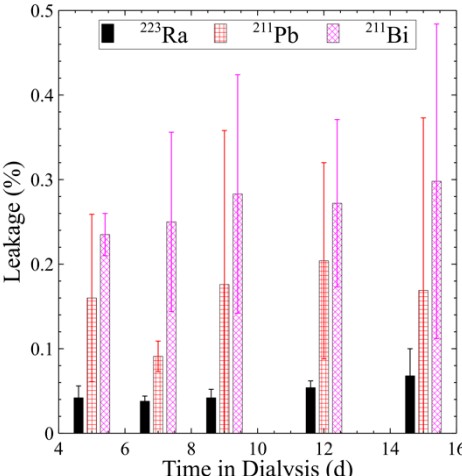

**Figure 8.** The retention of $^{223}$Ra and its decay daughters ($^{211}$Pb, $^{211}$Bi) within La($^{223}$Ra)VO$_4$ NPs is enhanced when the NPs are dialyzed against PBS compared to dialysis against deionized water. Radionuclide leakage of La($^{223}$Ra)VO$_4$ NPs (*n* = 3), synthesized by procedure C, was assessed after dialysis against PBS (1X, pH 7.2) to expose the particles to biologically relevant conditions.

La($^{223}$Ra)VO$_4$ and La($^{223}$Ra, $^{140}$Ba)VO$_4$ NPs have exhibited promising capabilities as delivery platforms for $^{223}$Ra, as well as alkaline and lanthanide radionuclides. The synthesis conditions of LaVO$_4$ NPs can be tailored to enhance the radiochemical yield and the retention of radionuclides within the NPs. It is expected that the fraction of $^{223}$Ra incorporated can be increased significantly (up to ~10 at. %) without affecting the crystal structure of LaVO$_4$ NPs [52]. Although the nature of functional groups (i.e., hydroxyl groups) limits the colloidal stability and functionalization of LaVO$_4$ NPs, the particle

surface can be modified with bifunctional compounds. The use of bifunctional compounds has the potential to increase the stability of LaVO$_4$ NP suspensions, prevent their accumulation in the RES, and allow their functionalization with targeting vectors. The selection of bifunctional compounds must be followed by an evaluation of the physicochemical characteristics of the modified NPs and an assessment of their encapsulation and retention of radionuclides.

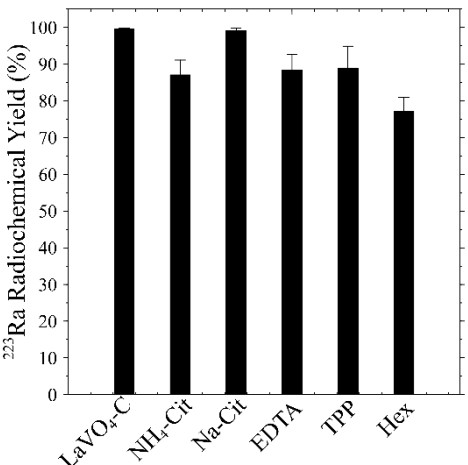

**Figure 9.** The $^{223}$Ra radiochemical yield decreased after surface modification with different stabilizing compounds. The effect of surface modification of La($^{223}$Ra)VO$_4$ NPs, synthesized using procedure C, with different stabilizing compounds was assessed based on their $^{223}$Ra radiochemical yield.

## 4. Conclusions

LaVO$_4$ NPs are promising delivery platforms for $\alpha$-emitting radionuclides based on the high encapsulation of $^{223}$Ra and the high retention of its decay daughters, $^{211}$Pb and $^{211}$Bi. Synthesis parameters, particularly the order of mixing the stock solutions and the final pH of the NP suspension, influenced the morphology, crystal structure, and radionuclide encapsulation and retention of La($^{223}$Ra, $^{140}$Ba)VO$_4$ NPs. Maintaining the La($^{223}$Ra, $^{140}$Ba)VO$_4$ NP suspension at an acidic pH (4.9) yielded a high fraction of NPs with tetragonal structure and resultant radiochemical yields of $^{223}$Ra and $^{140}$Ba <40%. Increasing the pH of the NP suspension to 11 resulted in the precipitation of a high fraction of La($^{223}$Ra, $^{140}$Ba)VO$_4$ NPs with a monoclinic structure and a twofold increase in $^{223}$Ra and $^{140}$Ba radiochemical yields (>85%). Surface modification of LaVO$_4$ NPs was indicated by the characteristic vibration bands associated with carboxylate and phosphate groups and resulted in a threefold decrease in hydrodynamic diameter and $^{223}$Ra radiochemical yields between 74.7% and 99.6%. Altering the surface of La($^{223}$Ra, $^{140}$Ba)VO$_4$ NPs enhances their colloidal stability and allows their functionalization with targeting vectors or fluorophores. The selection of bifunctional compounds must consider their effects on the encapsulation and retention of radionuclides over time. These results show the potential of La($^{223}$Ra)VO$_4$ NPs for TAT, where current efforts are focused on optimizing their surface chemistry for functionalization and evaluating their cytotoxicity using in vitro models.

**Supplementary Materials:** The following are available online at https://www.mdpi.com/2624-8 45X/2/1/3/s1, Figure S1: Decay schemes of $^{223}$Ra and $^{140}$Ba, Table S1: Summary of $\gamma$-ray energies and intensities used to calculate the activity of each radionuclide, Figure S2: FTIR spectra of LaVO$_4$ NPs, synthesized following procedure C (a) before and (b) after dispersion in PBS, Figure S3: FTIR spectra of LaVO$_4$ NPs modified with NH$_4$-Cit at different LaVO$_4$:NH$_4$-Cit molar ratios, Figure S4: FTIR spectra of LaVO$_4$ NPs modified with Na-Cit at different LaVO$_4$:Na-Cit molar ratios, Figure S5: FTIR spectra of LaVO$_4$ NPs modified with EDTA at different LaVO$_4$:EDTA molar ratios, Figure S6: FTIR spectra of LaVO$_4$ NPs modified with TPP at different LaVO$_4$:TPP molar ratios, Figure S7: FTIR spectra of LaVO$_4$ NPs modified with Hex at different LaVO$_4$:Hex molar ratios, Figure S8: FTIR

spectra of $LaVO_4 + NH_4$-Cit and $LaVO_4 + EDTA$ NPs (a, c) before and (b, d) after dispersion in PBS (1X, pH = 7.4).

**Author Contributions:** Conceptualization, M.T.-G., S.M., S.M.D.; methodology, M.T.-G., S.M.D.; radioactive materials preparation, A.P., R.C.; characterization, M.T.-G., A.M., D.A.C.; writing—original draft preparation M.T.-G., S.M., S.M.D.; writing—review and editing, M.T.-G., A.M., R.C., S.M.D. All authors have read and agreed to the published version of this manuscript.

**Funding:** This research was sponsored by the Laboratory Directed Research and Development Program of Oak Ridge National Laboratory, managed by UT-Battelle, LLC, for the US Department of Energy (DOE).

**Note:** This manuscript has been authored by UT-Battelle, LLC, under contract DE-AC05-00OR22725 with the US Department of Energy (DOE). The US government retains and the publisher, by accepting the article for publication, acknowledges that the US government retains a nonexclusive, paid-up, irrevocable, worldwide license to publish or reproduce the published form of this manuscript, or allow others to do so, for US government purposes. DOE will provide public access to these results of federally sponsored research in accordance with the DOE Public Access Plan (http://energy.gov/downloads/doe-public-access-plan).

**Data Availability Statement:** Data is contained within the article and supplementary material.

**Acknowledgments:** Nanomaterials characterization was conducted at the Center for Nanophase Materials Sciences, which is a DOE Office of Science User Facility. This research was supported by the US DOE Isotope Program, managed by the Office of Science for Nuclear Physics, and by the ORNL Laboratory Directed Research and Development program. The authors would like to thank the staff of the Bethel Valley Radiochemical Processing group at ORNL for their contributions in isotope production and purification.

**Conflicts of Interest:** The authors declare no conflict of interest.

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
