# Peer review of "Tailoring the Radionuclide Encapsulation and Surface Chemistry of La(223Ra)VO4 Nanoparticles for Targeted Alpha Therapy"

_jnt, doi:10.3390/jnt2010003_

Round 1

Reviewer 1 Report

The manuscript submitted for review is written at a very high level and reflects the hot topic issues of targeted alpha particle therapy. The task of finding suitable vectors for the targeted transport of α-decaying radionuclides, e.g. 223Ra, 225Ac, and their daughter products is not easy. According to the presented results, the authors faced a number of problems associated with the synthesis and subsequent stabilisation of nanoparticles, which could be suitable candidates in the future as drug delivery systems. I am appreciated that task of recoils effect was opened and discussed in the manuscript.

However, the question of stabilising nanoparticles remains open. The authors attempted size tuning of the prepared LaVO4 nanoparticles, but as the data suggest, there is still enough room for optimisation and research. Equally important is the question of 223Ra labelling. Therefore, I have a several question and comments on authors, which should be discussed/reflected in manuscript:

1) In the preparation methods, the authors use pH changes in order to prepare as small stable nanoparticles as possible in high radiochemical yield. Did the authors study the pH dependence during coprecipitation / synthesis of LaVO4 nanoparticles with 223Ra? Here it would be interesting to compare and supplement with data in the literature, especially the DOI publication: 10.1039 / C9RA08953E, which deals with the labeling of 223Ra on somewhat similar material. Has the effect of pH change on the size of the prepared nanoparticles been observed? Has the effect of temperature on the size of the prepared nanoparticles also been observed / studied? Hydrothermal preparation methods are known which are applied but very often lead to particles with a larger hydrodynamic radius.

2) The authors state that particle aggregation can be associated with the lack of functional groups on the particle surface. (page.9, line 319) This statement seems to me to be in conflict with previous experience, or I do not see any support for it in previous data. Although the number of functional groups is important, I believe that the main role here will be played by the nature of the interactions with respect to the functional groups, in the case of sorption processes or interactions of charged parts on the surface. It does not seem to me that there is enough evidence in the manuscript for the above statement. If the authors insist on this statement, it would be desirable to add other characteristics of the prepared materials. Characterisation of the material in terms of specific surface area, the number of functional groups determined by titration, as well as the optimisation of reaction conditions is key for successful labelling but also for the aggregation of materials.

3) The structural characterization of the prepared nanomaterials is sufficient. For FT-IR, it would be appropriate to list the crystal material that was used for the measurement. The spectral data show that a diamond ATR crystal was used. In this context, the acquisition conditions are not very successful for modified nanoparticles. Probably methods of specular reflection, especially grazing angle, would be more suitable for measuring thin layers on the surface (shell layer) of nanoparticles. ATR has a higher penetration ability and the penetration of radiation to the effective depth (de) provides more bands from the LaVO4 matrix than from the surface layer. The bands corresponding to the added stabilising agents (citric acid, EDTA, or sodium citric acid) then disappear and are not clearly visible. Some functional groups (COOH) are visible in the figures in the supplements as mere arms or low-intensity bands (1600 cm-1) or (1200 - 1000 cm-1).

However, the manuscript is very successful and I recommend it for acceptance after minor revision and answering questions.

Author Response

1) In the preparation methods, the authors use pH changes in order to prepare as small stable nanoparticles as possible in high radiochemical yield. Did the authors study the pH dependence during coprecipitation / synthesis of LaVO4 nanoparticles with 223Ra? Here it would be interesting to compare and supplement with data in the literature, especially the DOI publication: 10.1039 / C9RA08953E, which deals with the labeling of 223Ra on somewhat similar material. Has the effect of pH change on the size of the prepared nanoparticles been observed? Has the effect of temperature on the size of the prepared nanoparticles also been observed / studied? Hydrothermal preparation methods are known which are applied but very often lead to particles with a larger hydrodynamic radius.

Response: We appreciate the reviewer questions and comments. Although we did not study the effect of pH in this work, we recently assessed the effect of pH change on the size of gadolinium vanadate nanoparticles (doi: 10.1515/ract-2019-3206), which were prepared by the same coprecipitation method described herein. The influence of temperature on the particle size was not evaluated since the synthesis method does not require temperature input. To reflect on the effects of pH change on the size of lanthanide vanadate nanoparticles and the encapsulation of 223Ra, we have included the following changes to the manuscript.

Modifications

Line 138-139: The proposed modifications were motivated by our previous work with GdVO4 NPs [41].

Line 299-302: These results are consistent with the characteristic morphology obtained for GdVO4 NPs when the pH was adjusted to > 11, where both large trapezoidal particles and aggregates of small crystallites were observed [41].

Line 438-441: It is also expected that formation of different radionuclide species as a consequence of pH changes will promote the uptake of radionuclides by surface sorption [65]. The sorption of radionuclides onto the particle surface could explain the high leakage observed for both 223Ra and 140Ba (Fig. 6a).

Line 457-459: It is assumed that the dialysate activity of 223Ra and 140Ba may originate from radionuclides that were not coprecipitated within LaVO4 NPs but rather adsorbed onto the particle surface [65].

Line 490-492: It is hypothesized that the sorption and surface complexation of 223Ra and phosphate compounds in solution with LaVO4NPs decreases the radionuclide activity in the dialysate [24], [65].

2) The authors state that particle aggregation can be associated with the lack of functional groups on the particle surface. (page.9, line 319) This statement seems to me to be in conflict with previous experience, or I do not see any support for it in previous data. Although the number of functional groups is important, I believe that the main role here will be played by the nature of the interactions with respect to the functional groups, in the case of sorption processes or interactions of charged parts on the surface. It does not seem to me that there is enough evidence in the manuscript for the above statement. If the authors insist on this statement, it would be desirable to add other characteristics of the prepared materials. Characterisation of the material in terms of specific surface area, the number of functional groups determined by titration, as well as the optimisation of reaction conditions is key for successful labelling but also for the aggregation of materials.

Response: The statement regarding the relation between nanoparticle aggregation and the functional groups on the particle’s surface has been revised. We agree with the reviewer on the fact that the nature of interactions between the functional groups and the nanoparticles are key in their stability. Our characterization results suggest that lanthanum vanadate nanoparticles are covered with hydroxyl groups and carbonate species, which will determine the particle stability in different media. We demonstrated that after surface modification the particle stability increased by means of a decrease in hydrodynamic diameter. This increase in particle stability by surface modification shows the influence of inter-particle attractive forces. We will consider additional characterization of the particle surface after surface modification in our efforts to increase the particle stability and to label the nanoparticles with targeting moieties. The modifications made to the manuscript are summarized below.

Modifications

Line 326-327: Particle aggregation can be associated with hydrogen bonding between surface hydroxyl groups [40].

Line 344-346: The nature of the functional groups of LaVO4 NPs synthesized by the different procedures results in poor colloidal stability and limits their functionalization with targeting vectors.

Line 520-522: Although the nature of functional groups (i.e., hydroxyl groups) limits the colloidal stability and functionalization of LaVO4NPs, the particle surface can be modified with bifunctional compounds.

3) The structural characterization of the prepared nanomaterials is sufficient. For FT-IR, it would be appropriate to list the crystal material that was used for the measurement. The spectral data show that a diamond ATR crystal was used. In this context, the acquisition conditions are not very successful for modified nanoparticles. Probably methods of specular reflection, especially grazing angle, would be more suitable for measuring thin layers on the surface (shell layer) of nanoparticles. ATR has a higher penetration ability and the penetration of radiation to the effective depth (de) provides more bands from the LaVO4 matrix than from the surface layer. The bands corresponding to the added stabilising agents (citric acid, EDTA, or sodium citric acid) then disappear and are not clearly visible. Some functional groups (COOH) are visible in the figures in the supplements as mere arms or low-intensity bands (1600 cm-1) or (1200 - 1000 cm-1).

Response: We appreciate the suggestions made by the reviewer. The crystal material used for the FTIR measurements has been included. We will consider using the suggested characterization techniques to reflect the surface modification with each of the stabilizing agents and to determine the functional groups on the particle surface as part of our efforts on optimizing the surface chemistry and stability of lanthanum vanadate nanoparticles.

Modifications

Line 189-191: Surface characterization was carried out using attenuated total reflectance FTIR spectroscopy on a Nicolet™ iS50 FTIR spectrometer with a diamond crystal (Thermo Scien-tific™, MA).

Please see attachment for complete response letter.

Reviewer 2 Report

One approach to retaining the daughters from in-vivo alpha generators in TAT is that presented by the authors. In my opinion, the introduction would be strengthened by describing the alternative strategy of using biological vectors that are rapidly taken into the target cells. This approach has already demonstrated success in the use of PSMA agents to deliver 225Ac.

Given the audience of the journal, I recommend inserting "(stopping power)" after high-Z on line 66 of the introduction.

I recommend changing line 122 to "was obtained from an 227Ac generator; the parent was produced via thermal neutron . . ."

It is unclear to me why the radiochemical yield was determined by subtracting losses observed from so many materials. This approach increases the uncertainty on the reported value. Why was the RY% not measured by the difference between the initial activity and the product material from the synthesis?

Given the statement in lines 407 through 410, why not perform the synthesis under inert atmosphere (Ar or nitrogen) to avoid the formation of carbonate species?

Author Response

  • One approach to retaining the daughters from in-vivo alpha generators in TAT is that presented by the authors. In my opinion, the introduction would be strengthened by describing the alternative strategy of using biological vectors that are rapidly taken into the target cells. This approach has already demonstrated success in the use of PSMA agents to deliver 225Ac.

Response: We appreciate the suggestion of including alternative strategies for delivering alpha-emitting radionuclides. We have made the following modifications to the manuscript.

Modifications

Line 56-59: Although the relocation of decay daughters can lead to unwanted radiotoxicity to healthy organs and a decrease in therapeutic efficacy, radioimmunoconjugates with rapid uptake into cancer cells can overcome these limitations and have shown remarkable therapeutic efficacy [10]–[12].

  • Given the audience of the journal, I recommend inserting "(stopping power)" after high-Z on line 66 of the introduction.

Response: Per reviewer recommendation we have made the following changes.

Modifications

Line 67-69: An enhanced retention of decay daughters was obtained by precipitating 225Ac-InPO4 nanocrystals within polymersomes thanks to the high stopping power (i.e., high-Z) environment surrounding the α-emitting radionuclide [17].

Line 69-71: The principle of surrounding α-emitting radionuclides with a high stopping power environment has been used for developing inorganic NPs.

  • I recommend changing line 122 to "was obtained from an 227Ac generator; the parent was produced via thermal neutron . . ."

Response: We have modified the sentences related with the production/generation of 223Ra. The following modifications were made to the manuscript.

Modifications

Line 125-126: Carrier-free 223Ra was obtained from 227Ac generator, where the parent radionuclide was produced via thermal neutron irradiation of a 226Ra target.

  • It is unclear to me why the radiochemical yield was determined by subtracting losses observed from so many materials. This approach increases the uncertainty on the reported value. Why was the RY% not measured by the difference between the initial activity and the product material from the synthesis?

Response: We appreciate the reviewer’s recommendation of calculating/reporting the radiochemical yield as the difference/fraction of the initial activity and the activity on the final nanoparticle suspension. We calculated the radiochemical yield by subtracting losses from different materials/steps because we were interested in identifying the possible mechanisms/causes behind a low or high radiochemical yield. By taking into consideration each material/step we identified that some samples lost a high fraction of activity after the first dialysis/cleaning, suggesting that 223Ra was not incorporated within lanthanum vanadate nanoparticles. In other circumstances, the activity lost with time was associated with 223Ra adsorbed on the particle surface as evidenced by a continuous increase in dialysate activity with time. After comparing of both methods, the one reported in the manuscript and the one suggested by the reviewer, we did not notice a significant difference in the mean radiochemical yield and its uncertainty. See table below for comparison:

Procedure

223Ra RY (%) based on Eq (2)

223Ra RY (%) based on reviewer’s method

LaVO4-A

19.2 ± 4.1

20.9 ± 1.2

LaVO4-B

37.2 ± 6.9

37.6 ± 3.0

LaVO4-C

87.4 ± 6.9

91.7 ± 6.4

  • Given the statement in lines 407 through 410, why not perform the synthesis under inert atmosphere (Ar or nitrogen) to avoid the formation of carbonate species?

Response: We were not expecting the precipitation of carbonate species during the coprecipitation synthesis of lanthanum vanadate nanoparticles. We selected this coprecipitation method for its simplicity and high yield. We will consider the synthesis of nanoparticles under an inter atmosphere to prevent the formation of these species and assess its effects on the radiochemical yield and particle stability.

Please see the attachment for the complete response letter.
